# Optimizing Residential Construction Site Selection in Mountainous Regions Using Geospatial Data and eXplainable AI

Dhafer Alqahtani [1], Javed Mallick [1], Abdulmohsen M. Alqahtani [1] and Swapan Talukdar [2,*]

1   Department of Civil Engineering, College of Engineering, King Khalid University, Abha 61421, Saudi Arabia; dhafer@kku.edu.sa (D.A.); jmallick@kku.edu.sa (J.M.); hso0on-20@hotmail.com (A.M.A.)
2   Department of Geography, Asutosh College, University of Calcutta, Kolkata 700026, India
*   Correspondence: swapan.talukdar@asutoshcollege.in

**Abstract:** The rapid urbanization of Abha and its surrounding cities in Saudi Arabia's mountainous regions poses challenges for sustainable and secure development. This study aimed to identify suitable sites for eco-friendly and safe building complexes amidst complex geophysical, geoecological, and socio-economic factors, integrating natural hazards assessment and risk management. Employing the Fuzzy Analytic Hierarchy Process (Fuzzy-AHP), the study constructed a suitability model incorporating sixteen parameters. Additionally, a Deep Neural Network (DNN) based on eXplainable Artificial Intelligence (XAI) conducted sensitivity analyses to assess the parameters' influence on optimal location decision making. The results reveal slope as the most crucial parameter (22.90%), followed by altitude and land use/land cover (13.24%), emphasizing topography and environmental considerations. Drainage density (11.36%) and rainfall patterns (9.15%) are also significant for flood defense and water management. Only 12.21% of the study area is deemed "highly suitable", with "no-build zones" designated for safety and environmental protection. DNN-based XAI demonstrates the positive impact of variables like the NDVI and municipal solid waste generation on site selection, informing waste management and ecological preservation strategies. This integrated methodology provides actionable insights for sustainable and safe residential development in Abha, aiding informed decision making and balancing urban expansion with environmental conservation and hazard risk reduction.

**Keywords:** sustainable urbanization; GIS-based site selection; risk assessment; artificial intelligence; mountainous terrain; decision-making framework

## 1. Introduction

The Kingdom of Saudi Arabia (KSA), a major global oil exporter, has seen its construction sector grow due to population growth, which has increased the demand for new structures and resources [1–4]. The "Saudi Green Building Council (SGBC) 2010" advocates for sustainable building practices [5]. Site selection, which is critical in the design of buildings, directly impacts subsequent design decisions, which are guided by site suitability parameters based on the intended use. This highlights the central role of architects and engineers, who use their expertise and data to inform decision making [6–8]. Optimal site selection minimizes resource consumption and ensures logistical and economic feasibility, which is essential for project success [9]. Urban sustainability, with site selection as a starting point, considers environmental impact, energy consumption, safety, accessibility, and integration with existing infrastructure [10–14]. In regions such as Aseer, which are known for their mountainous terrain, land use, slope stability, drainage, and environmental impacts must be evaluated during site selection [15,16]. Geographic information systems (GISs) are critical for visualizing and analyzing these factors and aid in decision making, improving spatial navigation, infrastructure planning, and regional sustainability [17–19]. GISs also play a central role in creating and sharing 3D building models considering geographic attributes and are used in various projects, such as shopping malls, real estate

projects, and landscape impact assessments [20–23]. In addition, the integration of GISs with Building Information Modeling (BIM) improves the process of site selection [23].

The multi-criteria decision making (MCDM) approach, in particular the Analytical Hierarchy Process (AHP), is used to tackle complex decisions by assigning weights to thematic layers based on expert judgment and site-specific conditions, thus evaluating the relative importance of different parameters [24–26]. Despite its widespread application in geospatial zonation mapping, groundwater studies, and hazard mapping, the AHP faces significant challenges in dealing with imprecise input data and inherent uncertainties that can significantly affect the decision-making process [27–31]. These uncertainties can include the subjective judgments of experts, variability in data quality, and inconsistencies in the weighting of criteria, which can lead to less reliable results [27,32]. To improve accuracy and solve these problems, the integration of the AHP with fuzzy set theory, which forms the Fuzzy-AHP, has been proposed. This integration allows for a more nuanced consideration of uncertainties by incorporating linguistic variables and fuzzy numbers, thus refining accuracy in applications such as the selection of safe sites for residential building construction [28,29,33]. This study aims to develop integrated methods using remote sensing, geographic information systems (GISs), and Fuzzy-AHP for the selection of safe building sites in mountainous regions. The approach comprehensively considers environmental, safety, accessibility, and energy consumption factors and aims to evaluate the long-term integration of the project into the community and increase decision making reliability [24,34,35].

Recent research emphasizes the effectiveness of GIS-based multi-criteria decision making in mountainous regions but shows critical gaps in the consideration of dynamic socio-economic factors that are essential for sustainable urban development [36–38]. The integration of artificial intelligence (AI) techniques could close these gaps. Artificial neural networks (ANNs), for example, are promising for analyzing complex, nonlinear dependencies between socio-economic and environmental factors. These networks can process huge datasets typical of urban planning to detect patterns that may not be apparent using conventional methods, improving the accuracy and sustainability of building site selection [36]. In addition, the application of the fuzzy technique for Order of Preference by similarity to the ideal solution (fuzzy-TOPSIS) can overcome the ambiguity and uncertainty inherent in human judgments about site selection criteria. This method integrates both quantitative and qualitative data and provides a more holistic approach for evaluating sites based on multiple sustainability indicators [39]. Multi-agent systems with deep reinforcement learning (DRL) introduce adaptive learning capabilities that can continuously improve site selection strategies based on real-time data. These systems are particularly adept at optimizing complex decisions involving multiple stakeholders and different objectives, such as balancing environmental impacts and community needs [40–43]. By incorporating these advanced AI approaches, the methodology not only becomes more robust in dealing with the complexity of the decision environment but also considers a broader range of criteria, such as quality of life and population preferences [42]. This enables a more comprehensive evaluation of potential sites and ensures that decisions are sustainable and in line with long-term urban development goals [40,41].

To address the identified research gaps, this study sets a clear goal: to develop an integrated decision-making framework that improves the selection of safe, environmentally friendly housing complexes in mountainous regions. The framework innovatively extends traditional GIS-based approaches by incorporating socio-economic dimensions and employing advanced artificial intelligence algorithms, with the specific aim of optimizing sustainable development and minimizing environmental impacts. The novelty of this research lies in its methodological integration of socio-economic parameters with the Fuzzy-AHP, Deep Neural Network (DNN)-based eXplainable Artificial Intelligence (XAI), and game theory. This comprehensive approach aims to deepen our understanding of the variables that influence optimal site selection in mountainous regions. By synergistically merging these different elements, the proposed framework aims to fill the existing gaps in

the literature by taking into account socio-economic considerations as well as geophysical and geoecological aspects. Furthermore, the innovative application of DNN-based XAI and game theory is of particular importance as it not only increases the robustness of the analysis but also provides detailed insights into the complex relationships between the different decision factors. This approach enables a more nuanced and science-based decision-making process that better takes into account the complexity of the natural and urban environment in site selection.

## 2. Materials and Methods

### 2.1. Description of Study Area

This study focuses on four cities in the southwestern region of Saudi Arabia: Abha, Khamis Mushayet, Ahad Rufaida, and Wadiyan (Figure 1). Renowned for their rich biodiversity, these urban highlands are popular tourist destinations [44]. The study area, spanning 1291 km$^2$, is characterized by dominant species like "J. procera trees, *Acacia origena*, and *A. gerrardii*". The cities are situated between latitudes 18°9′33.126″ N and 18°30′56.566″ N and longitudes 42°23′52.477″ E and 42°51′42.832″ E, with an average elevation of 2102 m ranging from 1557 to 2743 m above sea level. The region experiences heavy rainfall from February to June, resulting in occasional flash floods in nearby villages. Over the past 55 years, annual rainfall has averaged 245 mm, primarily between February and June. The average high and low temperatures stand at 30.8 °C and 9.4 °C, respectively. The geology composition comprises upper "Proterozoic plutonic rocks, spanning from gabbro to granite, alongside upper Proterozoic metamorphosed volcanic and sedimentary rocks belonging to the Bahah and Jiddha groups" [45]. The area displays both natural geological erosion and human-induced land degradation.

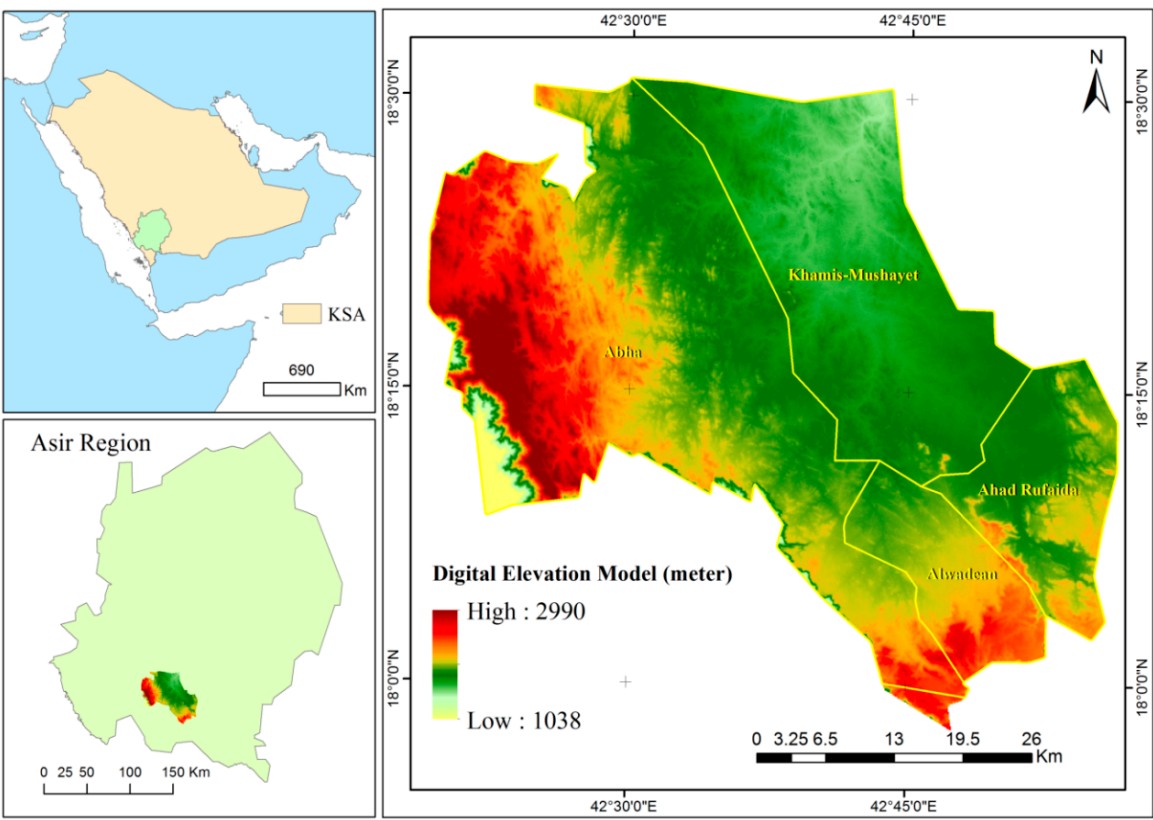

**Figure 1.** Location of the study area.

The region's diverse terrain experiences severe erosion due to slopes, weak geology, and heavy rain, impacting agriculture and forests. Residents rely on agriculture and secondary activities for their livelihoods, necessitating conservation and development efforts [46,47]. These characteristics make it an ideal case study location to illustrate the methodology's applicability. The Abha and Khamis Mushyet region in Aseer province, featuring the 3000-m-high Jabal Al-Sooda, boasts rich and varied flora [48]. Variations in environment and topography have fostered diverse plant communities. Juniperus procera dominates foggy, cold areas, while Acacia trees thrive in the western highlands. Ficus salicifolium communities and Ziziphus spina-christi var. spina-christi are common in lowlands, while various other species inhabit steep slopes to the west and south [49].

### 2.2. Data and Material Used

In this study, Sentinel datasets were utilized. Satellite data were utilized to generate maps depicting LULC, the NDVI, road networks, and various other features. Images from space: On 8 February 2020, Sentinel cloud-free data with a spatial resolution of 10 m were gathered from the archives of the Earth Explorer website. "The digital elevation model (DEM) ALOSPALSAR (https://asf.alaska.edu/data-sets/derived-data-sets/alos-palsar-rtc/alos-palsar-radiometric-terrain-correction/ (accessed on 11 March 2024)) at a resolution of 12.5 m was provided by NASA's Earth Science Data Systems". Between January 1 and 12 March 2023, extensive fieldwork and reconnaissance surveys were conducted across multiple locations to verify the diverse categories of land use and land cover (LULC) and to precisely locate the sites hosting ongoing construction for buildings. To process the data, ArcGIS 10.3, IDRISI, ERDAS, and SPSS software were used. In this research, Sentinel satellite datasets covering the study area were used. Below are the specifics of the satellite data that were used (Table 1).

**Table 1.** Data used for the present study.

| S. No | Data Type | Date of Production | Resolution | Source |
| --- | --- | --- | --- | --- |
| 1 | Sentinel satellite data | 2019-02-08 | 10, 20, 60 m | Earth Explorer (NASA) |
| 2 | Administrative map | 2005 | 1:50,000 | MOMRA, KSA |

### 2.3. Preparation of Factors for Suitable Site Selection for Residential Buildings

Several parameters were evaluated when selecting safe locations for residential building in the study area. Each parameter provided specific information related to safety considerations, including elevation, slope, aspect, vegetation density (NDVI), rainfall, land use and land cover (LULC), drainage density, distance to road networks, distance to industrial areas, distance to the airport, distance to academic institutions, distance to municipal waste, distance to shopping centers, distance to health services, distance to sewage treatment plants, restricted areas, and geology (Figures 2 and 3). Geological environmental conditions, geophysical characteristics, and socio-economic aspects are important elements to consider when selecting a site for residential development to ensure safety. Specific information regarding these elements is presented below:

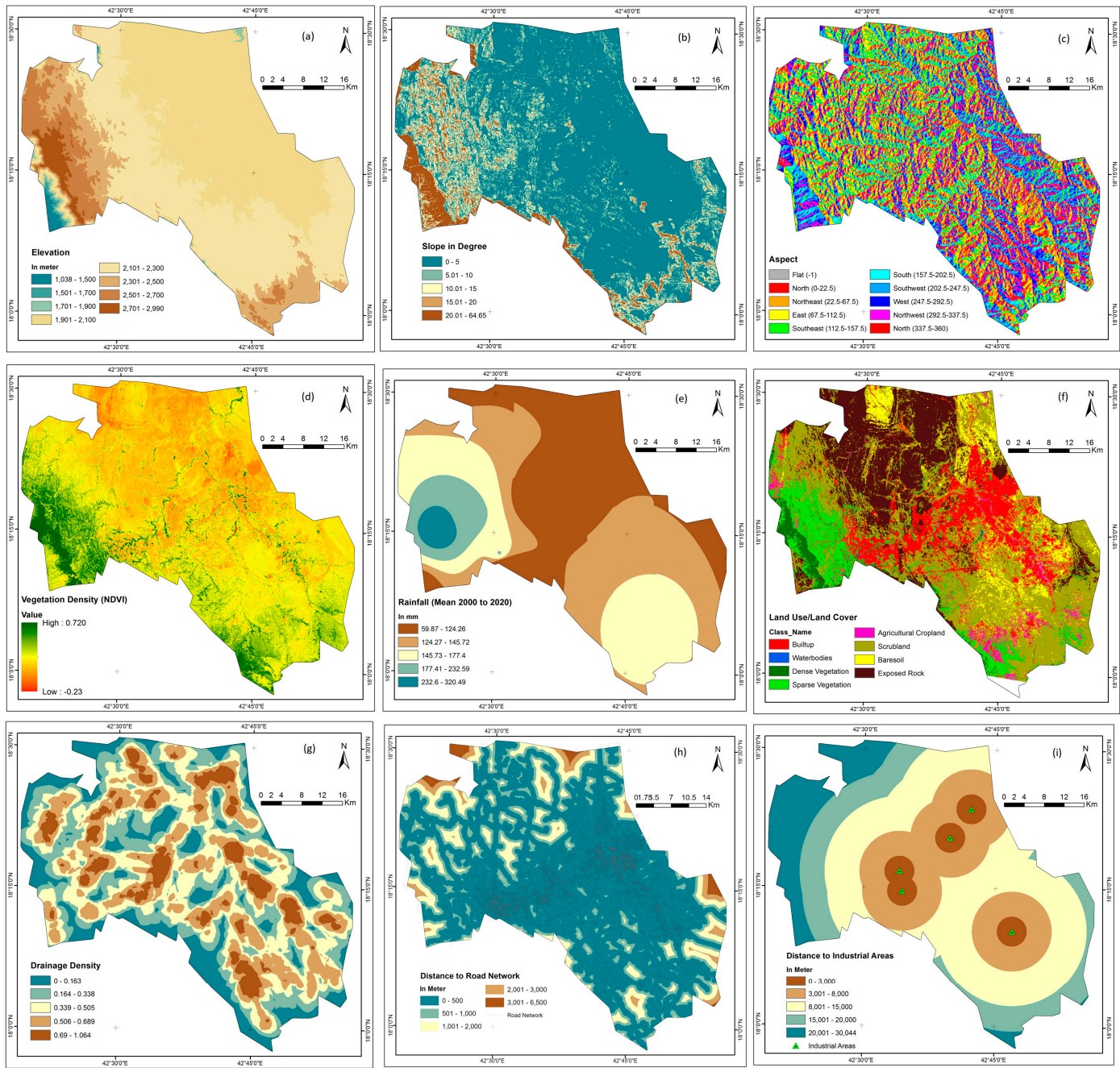

**Figure 2.** Thematic layers (such as (**a**) Elevation, (**b**) slope, (**c**) aspect, (**d**) vegetation density, (**e**) rainfall, (**f**) LULC, (**g**) drainage density, (**h**) distance to road network, (**i**) distance to industrial) for safe residential construction site selection.

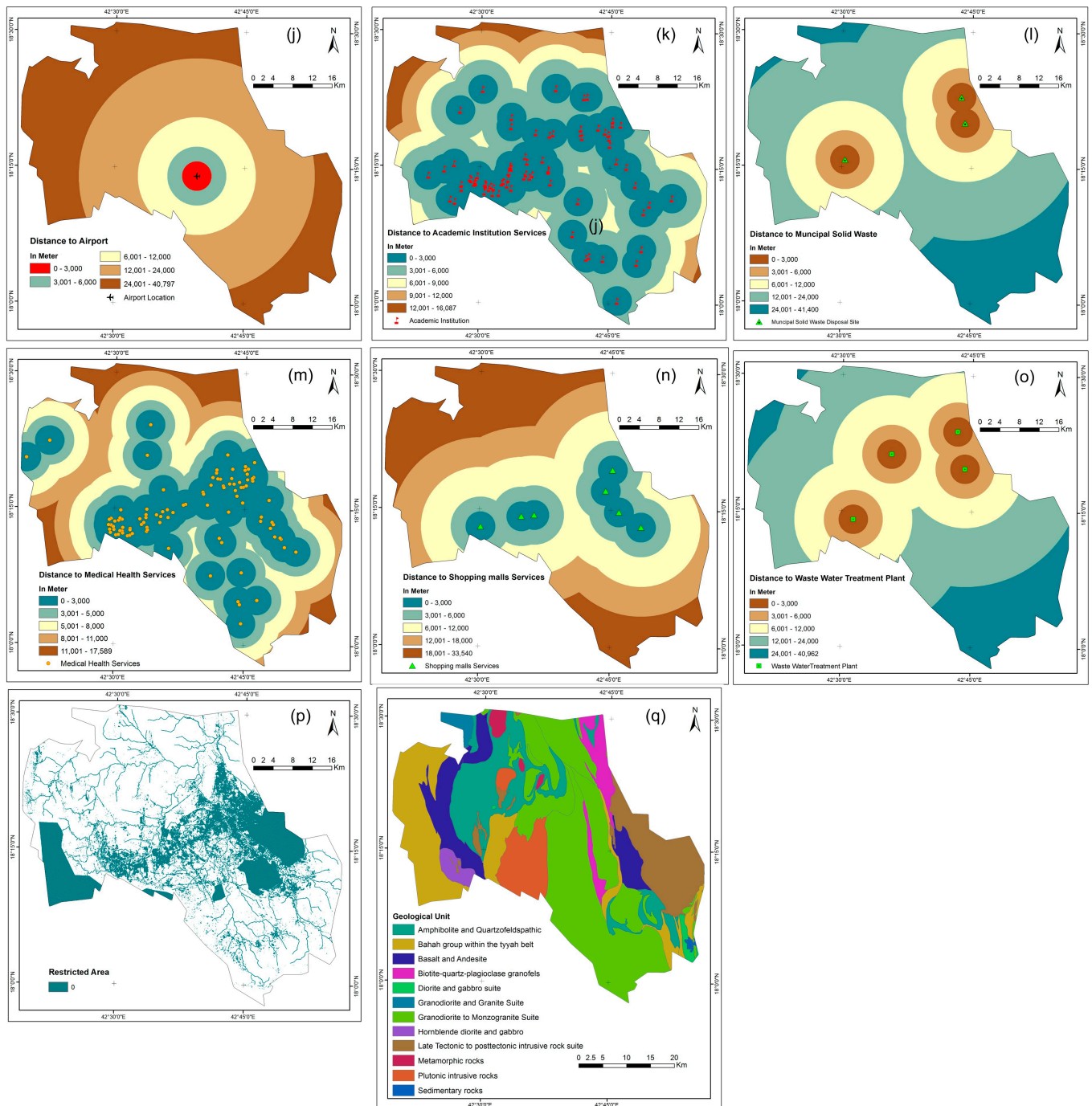

**Figure 3.** Thematic layers (such as such as (**j**) distance to airport, (**k**) proximity to academic institutions, (**l**) distance to MSW, (**m**) distance to health services, (**n**) distance shopping malls, (**o**) distance to WWTP, (**p**) Restricted area (**q**) geology) for safe residential construction site selection.

### 2.3.1. Elevation

The elevation of the study area ranges from 1038 to 2990 m above sea level, with an average elevation of 2179 m and a standard deviation of 207.49 m. The western part, which corresponds to the mountainous area of Alsooda, is higher than the east. The western region has a very high altitude, ranging from 2501 to 2990 m above mean sea level (AMSL). The center and west central areas have a moderate elevation, ranging from 2101 to 2500 m above sea level. Approximately 41% of the region, which lies between 1901 and 2100 m above mean sea level (AMSL), consists of low-lying areas that are ideal for safe residential building. Rainwater harvesting reservoirs are more suitable for lower elevations, while

elevated reservoirs are more suitable for higher elevations. Elevation data, along with slope and aspect calculations, are critical to assessing site safety for residential construction, as noted in expert interviews.

### 2.3.2. Slope

The study area has a slope that varies between 0 and 61.970 degrees, with an average of 6.060 degrees (standard deviation 5.660). About 50% of the area has a gentle slope of less than 10 degrees, while the western and central parts have a steeper slope. The slopes in the eastern, northeastern, and southeastern sections are less steep. Abrupt changes in slope can have detrimental effects on the ecosystem and lead to instability in the adjacent areas. Construction on slopes steeper than 30% or in landslide-prone areas is prohibited under the International Building Code (IBC).

### 2.3.3. Aspect

For residential development in hilly areas, slopes that receive a lot of sun are often preferred. South-facing slopes can absorb more heat compared to north-facing slopes, resulting in warmer environmental conditions [45]. Similarly, east-facing slopes receive sun in the morning, while west-facing slopes receive heat in the afternoon. Therefore, slopes facing south and east are important [49]. The score was influenced by the aspect map, which gives preference to slopes facing south and east, while slopes facing north and west are given less importance.

### 2.3.4. Measuring Vegetation Density with the Normalized Difference Vegetation Index (NDVI)

The preservation of ecosystems and resources is crucial for sustainable development. Urban growth in environmentally fragile areas should prioritize environmental protection, i.e., the protection of local flora and fauna. In order to reduce negative effects on the environment, it is important to protect forests, green spaces, and limited regions when constructing residential buildings. The vegetation in the study area, especially in the western region, is biologically diverse and must be protected from development. Adequacy ratings were assigned based on the NDVI values using the NDVI map (Figure 2). NDVI scores below 0.2 indicate probable adequacy, while scores above 0.2 indicate reduced adequacy.

### 2.3.5. Precipitation

Precipitation has a significant impact on residential site selection and affects the design, durability, and legal responsibility of structures. Consider the following precipitation criteria: 1. Historical data: analyze average annual rainfall, rainfall patterns throughout the year, and storm occurrence to evaluate the impact on the site. 2. Assess natural drainage patterns, proximity to flood prone areas and implement appropriate drainage systems to mitigate the risk of waterlogging and flooding. 3. Soil stability: Assess the ability of the soil to withstand rainfall and minimize foundation problems caused by clay expansion. Reduce susceptibility to erosion. 4. The presence of gentle slopes reduces the likelihood of landslides and erosion during periods of heavy rainfall. 5. Vegetation and permeability: lush vegetation is able to absorb rainfall, while permeable surfaces effectively reduce the amount of runoff, thus reducing the risk of flooding and erosion. 6. Capture rainwater: improve water conservation by choosing places with abundant rainfall for collection. 7. Comply with municipal regulations on rainwater and stormwater management. Adherence to these criteria will ensure the creation of safe, durable, and beneficial habitats for the future.

### 2.3.6. Land Use and Land Cover (LULC)

The land use and land cover (LULC) map for 2020 was created using the Maximum Likelihood Classification (MLC) algorithm for the reflectance bands to improve the distinction between the different land cover classes. The validation process, carried out using a confusion matrix, showed a high level of agreement, with an accuracy of 94.28% and a

kappa value of 0.927. Table S1 shows that 36.85% of the area comprises scrubland, while rocky terrain accounts for 23.65%. The percentage of water bodies is 0.02%, while the percentage of built-up areas is 14.86%, indicating a significant expansion of urban development. The percentage of land used for agriculture is 2.96%, mainly in rural areas. A total of 2.39% of the area is covered by dense vegetation, while 11.09% is covered by sparse vegetation. Water bodies, arable land, cultivated land, and dense and sparse vegetation are given a low weight, while scrubland, bare soil, and exposed rocks are given a high weight.

### 2.3.7. Density of Drainage

Over a 30-year period (1992–2022), the region received an average annual precipitation of 248 mm, with high-rainfall periods occurring from February to June. This heavy rain often leads to significant rainfall and sudden flooding in nearby rural areas [50]. To ensure safety, it is important that building sites are located at a considerable distance from drainage systems such as wadis (lakes, ponds, streams) to avoid the possible contamination of water sources. The assessment of the drainage network is of the utmost importance. The drainage density is determined by dividing the study area by the total number of wadis [51]. The optimal site for building a safe structure is at a distance from the drainage system, as shown in the second reference point.

### 2.3.8. Distance to Road Networks

The construction of new roads in hilly regions is associated with high costs. Therefore, efforts are concentrated on the proximity to existing routes. Buffer zones at a distance of 100 m from important roads improve accessibility. Transportation costs increase with increasing distance from the road, which affects adequacy. Suitability ratings decrease with increasing distance from the buffer zone, especially beyond a distance of 5 km. Immediate proximity to major highways, namely within 100 m, is rated extremely favorable.

### 2.3.9. Distance to Industrial Areas

A distance between industrial areas below 3000 m is not suitable for the construction of an industrial area. However, the lessened influence of industrial waste in these zones allows for the construction of residential areas above 6000 m. Therefore, experts suggest that buildings below 3000 m from industrial locations will receive less weight, and vice versa.

### 2.3.10. Distance to the Airport

It is advisable for residential complexes to maintain a minimum distance of 3 km from airports for safety reasons [45]. Buffer zones around airports are designated as exclusion zones to mitigate the effects of noise, landing patterns, and radar conflicts. The first checkpoint, which is at a distance of 3000 m, is not acceptable (Figure 3). However, the second checkpoint, which is 3000–6000 m away, is considered excellent. The third checkpoint, which gradually increases the distance from the airport, is also becoming unsuitable due to the longer travel time required to reach the airport.

### 2.3.11. Measuring Proximity to Academic Institution Services

When selecting locations for residential complexes, proximity to educational institutions such as colleges, schools, and universities is considered. Proximity to facilities is desirable for better accessibility. Euclidean distance is used for site evaluation. The first control point, located at a distance of 3000 m, is considered excellent. However, the second control point, located 6000 m away, is considered less acceptable. Residential complexes located in close proximity to facilities are theoretically considered more optimal (Figure 3).

### 2.3.12. Distance from Municipal Solid Waste

To ensure the safety of the construction sites of residential complexes, locations designated for the disposal of municipal solid waste must be avoided due to the associated health risks. To avoid these hazardous zones, buffer zones were defined using Euclidean

distance measurements. The resulting map was then classified and evaluated based on these buffer distances [45]. The second control point, which is well away from the municipal landfill, proves to be the first choice for selecting safe sites for the construction of residential complexes. In contrast, the first control point, which is 3000 m away, is a less favorable option for the construction of residential complexes.

### 2.3.13. Measuring Proximity to Medical Health Services

Healthcare is another important aspect when choosing the optimal site for a residential complex. There is a buffer zone around medical care facilities. The Euclidean distance was used to calculate the direct distances between healthcare facilities and possible locations for residential complexes. The wide location of the second control point, which is far away from medical healthcare facilities, shows that it is not suitable for the development of a residential complex. On the other hand, the first control point, which is close to healthcare facilities (within a radius of 3000 m), is the best location for building. Figure 3 illustrates the distribution of medical services in Abha and neighboring towns. Residential complexes are considered more advantageous if they are in close proximity to medical and health services than if they are further away.

### 2.3.14. Measuring Proximity to Shopping Center Services

Proximity to shopping centers was also taken into account when selecting locations for the residential complexes. People prefer shopping centers that are in close proximity to their homes. The buffer zones were created on the basis of Euclidean distance. The first control point, which is 3000 m closer to the shopping malls, represents an ideal location for building and shows potential. The second control point, which is further away from the shopping malls, indicates a less desirable site for construction. Most of the shopping malls are located in the southern and southeastern regions of the study area. Residential complexes are more desirable when they are closer to shopping malls, while distant locations are less suitable.

### 2.3.15. Distance to Sewage Treatment Plants

To ensure the safety and well-being of residents, it is important to carefully select the site for the building of residential complexes. It is recommended to keep a minimum distance of 3 km from wastewater treatment plants [45]. The diagram shows the spatial arrangement of these facilities and their respective Euclidean distances. Buffer zones were created around the locations of the wastewater treatment plants (WWTPs). The map was divided into segments and then evaluated based on the buffer distances. The first control point, located at a distance of 3000 m, represents an unfavorable site for the development of a residential complex. In contrast, the second control point, which is located far away from the region of the wastewater treatment plant, indicates the optimal distance for choosing a safe location for the construction of residential complexes, with the welfare of future residents as a priority.

### 2.3.16. Areas with Restricted Access

Maps showing protected or restricted regions were used together with an assessment of current built-up areas as an additional factor in the safety assessment. The restricted area also included a restricted zone, civil defense measures, and significant historical sites. Buffer zones included designated and restricted regions, and sites with a distance of more than 500 m were considered sufficient.

### 2.3.17. Geology

Geological characteristics play a crucial role in assessing the feasibility of a building site and ensuring its stability and safety [52]. Important aspects to consider are the properties of the soil, the depth of the bedrock, the amount of groundwater, the stability of the slope, the occurrence of seismic activity, the potential for erosion, and the presence of geological hazards. Engineers analyze these factors to select safe locations and develop strategies for foundation design, excavation, and mitigation. Geologic maps, such as the GM-75

Abha 1:250,000 Quadrangle, provide important geographic information, helping to make informed decisions.

*2.4. Method for Modeling Safe and Eco-Friendly Residential Construction Sites*

2.4.1. MCDM: Fuzzy Set Theory

Fuzzy set theory, developed by Lotfi A. Zadeh, extends classical set theory to manage partial membership and thus model the uncertainty prevalent in real-world scenarios, particularly in environmental and construction settings [53]. By enabling partial membership, this theory allows for more nuanced assessments compared to classical binary logic, which aligns well with the complexities of identifying eco-friendly and safe construction sites [54]. This flexible and adaptive approach is crucial for managing the imprecision often found in environmental data and stakeholder inputs, which are vital for planning sustainable urban developments [55].

As key elements, triangular fuzzy numbers (TFNs) are defined by parameters (l, m, u) representing the lowest possible value, the modal value, and the highest possible value, respectively. The membership function for a TFN is expressed as follows:

$$\mu\left(x \backslash \tilde{M}\right) = \begin{cases} 0, & x < l, \\ (x-l)/m-1), & l \le x \le m \\ (u-x)/(u-m), & m \le x \le u \\ 0, & x > u \end{cases} \tag{1}$$

Based on its left and right representation, the following equation displays the fuzzy number for each membership level [53]:

$$\tilde{M} = \left(M^{l(y)}, M^{r(y)}\right) = (l + m - l)y, u + (m - u)y) \tag{2}$$

where l(y) and r(y) represent a fuzzy number's left and right sides, respectively.

Fuzzy Membership Functions (FMFs) are an essential part of this theory and allow for the membership of elements to vary within a set. These functions, which can be triangular, trapezoidal, Gaussian, or sigmoidal, assign a degree of membership to each element that reflects the extent of its correspondence to a fuzzy set [56,57]. This method is crucial for decision-making processes where data uncertainty is prevalent, as it provides a structured approach for dealing with ambiguous information.

To standardize feature data, FMFs normalize the degree of membership of each data point from 0 to 1, facilitating comparisons between different scales and criteria. This normalization is crucial for spatial analysis and multi-criteria decision making. It helps in evaluating factors such as slope, vegetation, and proximity to infrastructures using different FMF types [58]. The Fuzzy Analytic Hierarchy Process (Fuzzy-AHP) is used to assign weights to these features, integrating expert opinion and fuzzy logic to refine decision accuracy.

In determining the weights, pairwise comparison matrices are created within a hierarchical structure, and fuzzy numbers are used to calculate the relative importance of each criterion. This process, detailed by equations from Mallick et al. [59], uses Buckley's geometric mean method to calculate geometric fuzzy means and weights [60,61], which are represented as triangular fuzzy numbers (TFNs) for each criterion, indicating lower, middle, and upper weighting values.

By incorporating fuzzy set theory and FMFs, this comprehensive modeling technique improves the precision and reliability of selecting safe and environmentally friendly construction sites by effectively managing the inherent uncertainties in environmental and construction planning.

The integration of feature thematic maps into a GIS environment is used to facilitate the residential safe site potential index (RSSPI) of the research area. Equation (3) and the "weighted linear combination (WLC) aggregation method" were used to calculate the RSSSZ [51], as shown below:

$$\text{RSSPI} = \sum_{w-1}^{m} \sum_{i=1}^{n} \left( wt_j \times x_i \right) \tag{3}$$

where RSSPI = the Residential Safe Site Potential Index, $x_i$ = thematic maps (FMFs), $wt_j$ = the normalized weight of the *jth* theme, m = the total number of themes, and n = the total number of classes in a theme.

2.4.2. Self-Organizing Maps (SOMs) for Classifying the Potential Residential Safe Site Map

Self-organizing maps (SOMs), introduced by Teuvo Kohonen, are pivotal for analyzing and classifying geographical data to determine potential sites for safe residential development. By reducing high-dimensional data into a two-dimensional, interpretable format, SOMs facilitate the visualization of complex spatial relationships and patterns that are critical for site selection. This method supports the decision-making process by providing a clear, organized representation of safety and environmental factors, which is essential for stakeholders aiming to balance development with ecological preservation.

However, an SOM is a form of artificial neural network. It transforms high-dimensional data into a reduced space while preserving data relationships. SOMs employ a 2D grid of neurons, each representing a cluster of similar data. Training involves iteratively presenting data and activating the nearest neuron, adjusting its weights to match the data. Once trained, new data are classified by finding the closest neuron, termed the best matching unit (BMU). Applications encompass data clustering, high-dimensional data visualization, and feature extraction. SOMs provide a user-friendly and versatile solution for diverse data analysis and visualization challenges.

*2.5. Process of Modeling SOMs*

To automate the classification of a single-band raster image using the k-means clustering algorithm and Self-Organizing Maps (SOMs), follow these steps: Begin by loading the single-band raster image as a point shapefile using software like ArcGIS, ensuring the shapefile includes X and Y coordinates for each point and a column for pixel values. Then, execute k-means clustering by loading the point shapefile through the geopandas library, extracting coordinates and values, converting data to a NumPy array, iterating through cluster numbers with sklearn.cluster.KMeans to compute silhouette scores using silhouette_samples from sklearn.metrics, and visualizing scores to determine the optimal cluster count. Normalize the data to a 0-1 scale, and proceed to create and train the SOM. Calculate the grid size based on clusters, instantiate minisom.MiniSom with the appropriate parameters, initialize weights using normalized data, and train the SOM. Assign cluster labels by obtaining cluster assignments with the labels_map method, and finally, generate a new shapefile containing clustered data using a GeoDataFrame with point coordinates and labels, before saving it to a desired location.

*2.6. Sensitivity Analysis as a Technique of eXplainable Artificial Intelligence through Deep Neural Networks*

Deep Neural Networks (DNNs), coupled with sensitivity analysis, offer a powerful approach to identifying and prioritizing factors that influence the selection of residential construction sites. This combination is particularly effective in environments where the impact of various criteria on the overall suitability of a site needs to be clearly understood and communicated to decision makers. Sensitivity analysis enhances the transparency and interpretability of DNN outputs, making it a valuable tool for explaining complex model decisions to stakeholders involved in planning eco-friendly and safe residential areas. This method ensures that critical variables influencing site selection are rigorously assessed and that the resulting decisions are robust and justifiable.

Sensitivity analysis is a technique used to assess the importance of input variables in a model and understand how changes in those variables affect the output. In this study, DNN-based sensitivity analysis was performed using the "NeuralSens" package in R. The DNN model was trained on the provided dataset, which includes variables such as LULC

(land use/land cover); distance from institutions, industrial areas, WWTPs (wastewater treatment plants), hospitals; slope; geology; distance from roads; a DEM (Digital Elevation Model); rainfall; the DD (Drought Index); the NDVI (Normalized Difference Vegetation Index); aspect; and distance from MSW (municipal solid waste), the airport, and malls. The DNN model was trained using the Keras library in R. It is a sequential model with multiple dense layers. The "relu" activation function is used for the hidden layers, and the final layer uses a linear activation function. The model was optimized using the RMSprop optimizer and the mean squared error loss function.

This study's method for modeling safe site selection is illustrated in Figure 4.

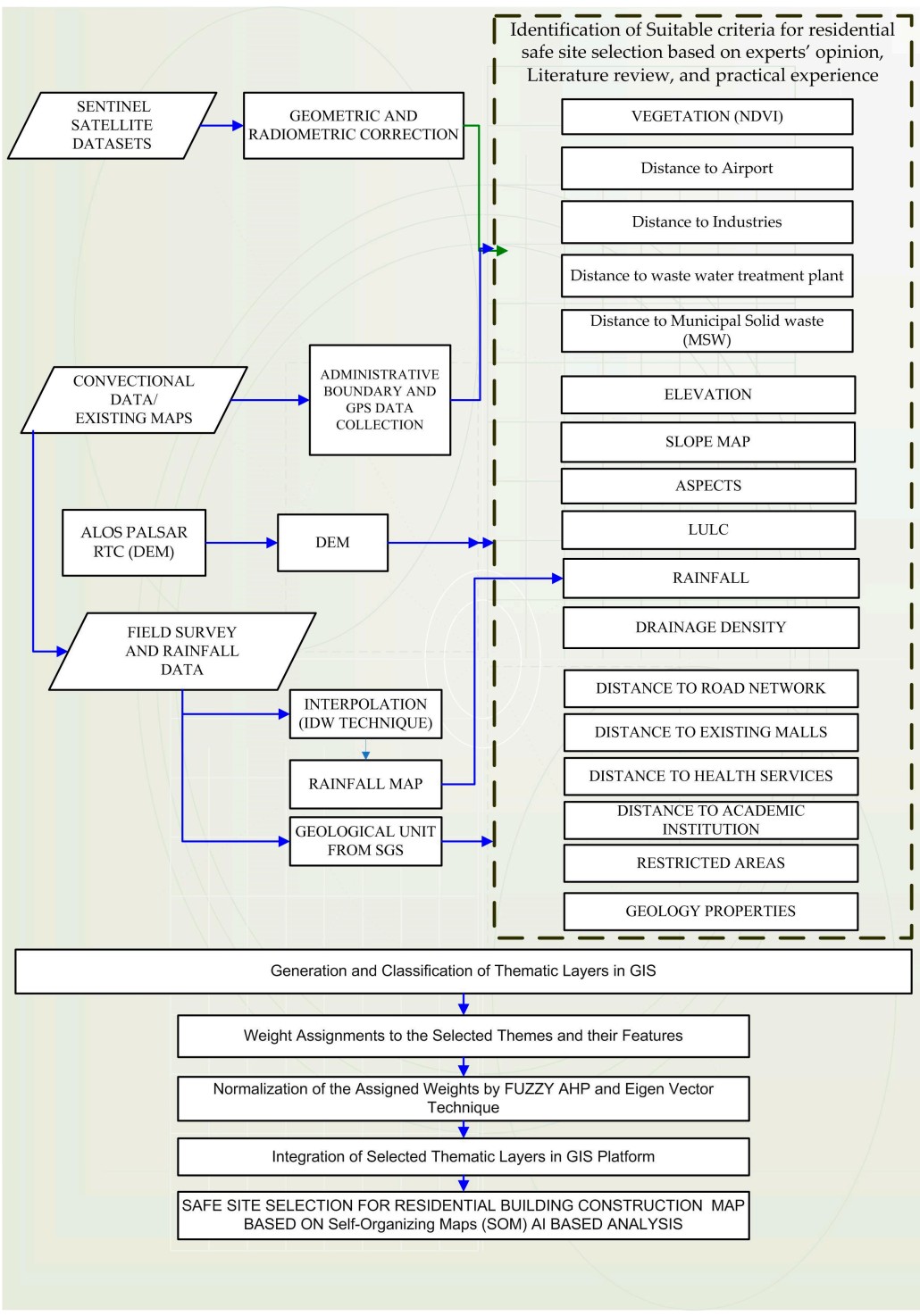

**Figure 4.** The general approach for the methodology.

## 3. Results

### 3.1. Normalization of Weights for Thematic Maps of Geophysical Aspects for Safe Site Selection Using Fuzzy-AHP

Geophysical, geoecological, and socio-economic parameters play an important role in determining the best location for the construction of residential complexes in mountainous terrains. Slope, elevation, drainage density, LULC, rainfall, vegetation density, geology, orientation, distance to road networks, distance to healthcare, distance to industry, distance to sewage treatment plant, distance to institutions, distance to shopping centers, and distance to the airport are just some of the integrated geophysical, geoenvironmental, and socio-economic factors that need to be considered when selecting a site. These factors can all be managed efficiently. According to the results of the study, weighted overlay analysis is the most effective way to map potential safe sites using the GIS-based Fuzzy-AHP. Using the Fuzzy-AHP method, a weight was assigned to each topic and class. The importance of each topic in the matrix was assessed based on a literature review, expert opinion, and field knowledge (Tables S2–S5). The most influential variable is the slope of the terrain, characterized by "FMS_Slope_linear30.img", with a weighting of 22.90%. This indicates that the slope of the terrain is of the greatest importance for both safety and environmental sustainability. This is closely followed by the variables "FMS_DEM_linearFinal.img" and "FMS_LULC_Catagories.img", both with a weighting of 13.24%, which emphasizes the importance of height and land use/land cover. Drainage density ("FMS_DD_linear.img") and rainfall patterns ("FMS_Rainfall_linear.img") also have a significant weight of 11.36% and 9.15%, respectively, indicating their role in flood prevention and water management. Vegetation, indicated by the NDVI ("FMS_NDVI_linear.img"), has a weight of 8.82%, emphasizing its role in preventing soil erosion and promoting local biodiversity. Interestingly, social infrastructure such as roads, hospitals, and sewage treatment plants have a lower weighting of 2.50% to 2.86%, suggesting that although they are important, they are secondary compared to environmental and topographical considerations. The variables with the lowest influence include aspects such as proximity to shopping centers ("FMS_Malls_linear.img") and airports ("FMS_Airport_linear.img"), with weightings below 2%. Consistency rates of 9.5%, delta = $8 \times 10^3$ a principal eigenvalue of 18.283, and the eigenvector solution are examples of the statistical analysis (Figure 5). The five iterations of the weights applied to the sixteen thematic levels and their attributes show that they are appropriate for the expected results.

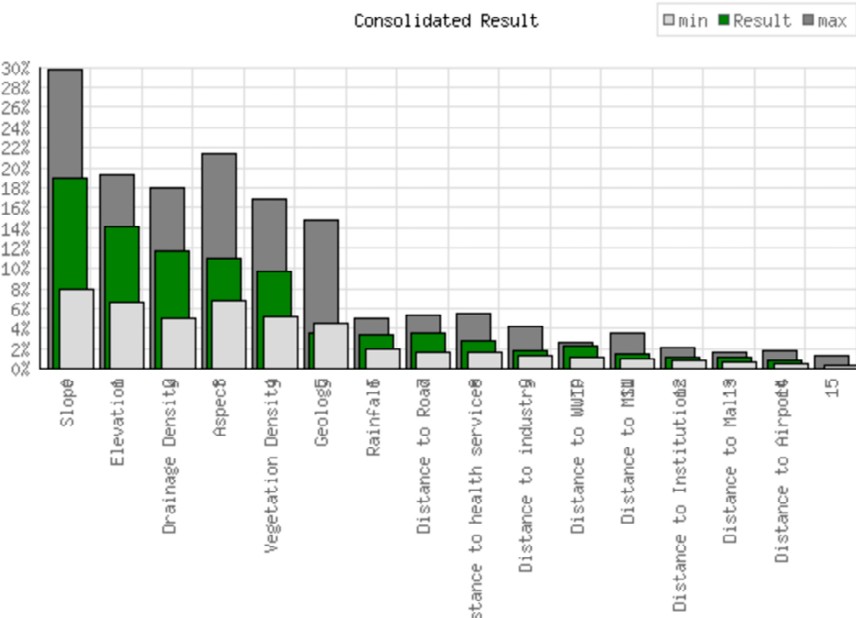

**Figure 5.** Computed weights are based on the principal eigenvector of the decision matrix.

Strategically, these results suggest that a holistic approach that prioritizes topographical and environmental considerations would be most effective. Special focus should be given to analyzing the slope and elevation of the prospective construction sites, followed by a comprehensive study of the land use and land cover to ensure minimal environmental impact. Incorporating these high-weight variables into the planning and decision-making process can significantly contribute to the safety and ecological sustainability of the new building complex.

*3.2. Generation of Safe Site Selection for Safe and Eco-Friendly Residential Complexes Using SOMs*

After the normalization of the sixteen thematic layers using the Fuzzy-AHP, all parameters can be characterized as fuzzy crisp layers, having values ranging from 0 to 1 in stretch form (Figure 6). Weighted overlay analysis was employed to integrate the sixteen parameters in a GIS (ArcGIS version 10.8) to generate possible safe sites. After integration in the GIS (Version 10.8), the RSSPI is generated, and the results are displayed in Figure 7. The RSSPI underwent zoning through histogram profiling using an SOM. The resulting zones exhibited a mean of 0.491 and a std. dev. of 0.299, with min and max values ranging from 0.0 to 0.793. Secure locations were identified and classified into six discrete categories based on specific criteria and characteristics, including areas designated as "Restricted" for no construction, as well as those classified as "Very Low", "Moderate", "High", and "Very High" areas deemed appropriate for the construction of residential buildings. Table 2 offers comprehensive statistics detailing the integrated categories of safe sites for residential building construction. According to the statistical study, 12.21% and 34.34% of the entire area was classified as extremely high and high SSSZ, respectively, while 15.18% and 7.86% was considered moderate and low appropriate potential. In various zones spanning from central–north to central–west and extending towards the eastern regions of studied area, there are identified zones highly suitable for safe residential building construction, with some areas classified as very high-suitability zones.

**Table 2.** Safe Site selection statistics for residential construction in Abha and surrounding cities.

| Class | Area (km) | % |
|---|---|---|
| Very High-Suitability | 279.23 | 12.21 |
| High-Suitability | 785.576 | 34.34 |
| Moderate-Suitability | 347.17 | 15.18 |
| Low-Suitability | 179.71 | 7.86 |
| Very Low-Suitability | 79.82 | 3.49 |
| Restricted Area | 616.21 | 26.94 |
| Total | 2287.70 | 100.00 |

In the analysis, the optimal number of clusters for the SOM was set at five based on the elbow diagram and the silhouette scores. The highest silhouette score, 0.33, was observed with two clusters, while a five-cluster configuration resulted in a score of 0.11. The SOM was configured with a grid size corresponding to five clusters, a sigma value of 0.50, and a learning rate of 0.50. During training, the SOM effectively learned data patterns and allowed cluster labels to be assigned based on the grid position of each data point. These labels facilitated the classification of points into their respective clusters, which were then geographically represented in a shapefile. This shapefile added a new field displaying the cluster assignments for further analysis. For visual representation, the shapefile can be converted into a raster image using GIS tools that assign pixel values based on cluster labels, making data analysis and visualization even easier.

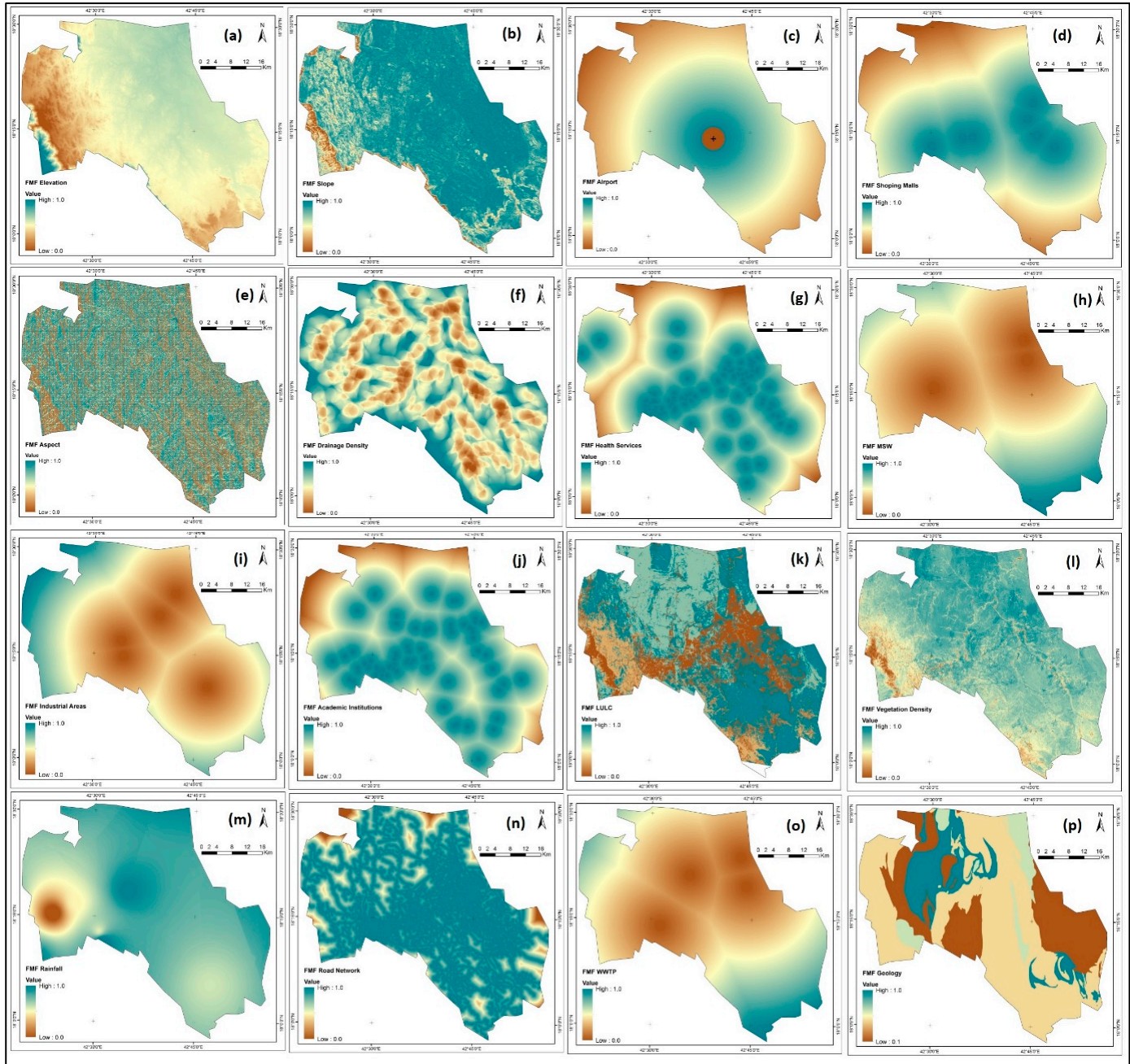

**Figure 6.** Fuzzified thematic layers, such as (**a**) Elevation, (**b**) slope, (**c**) proximity to airport, (**d**) proximity to shopping mall, (**e**) Aspect, (**f**) Drainage density, (**g**) proximity to health services, (**h**) MSW, (**i**) proximity to industrial areas, (**j**) proximity to academic institutions, (**k**) LULC, (**l**) vegetation density, (**m**) rainfall, (**n**) road network, (**o**) WWTP, (**p**) geology for generating RSSSI map with value ranges from 0 to 1.

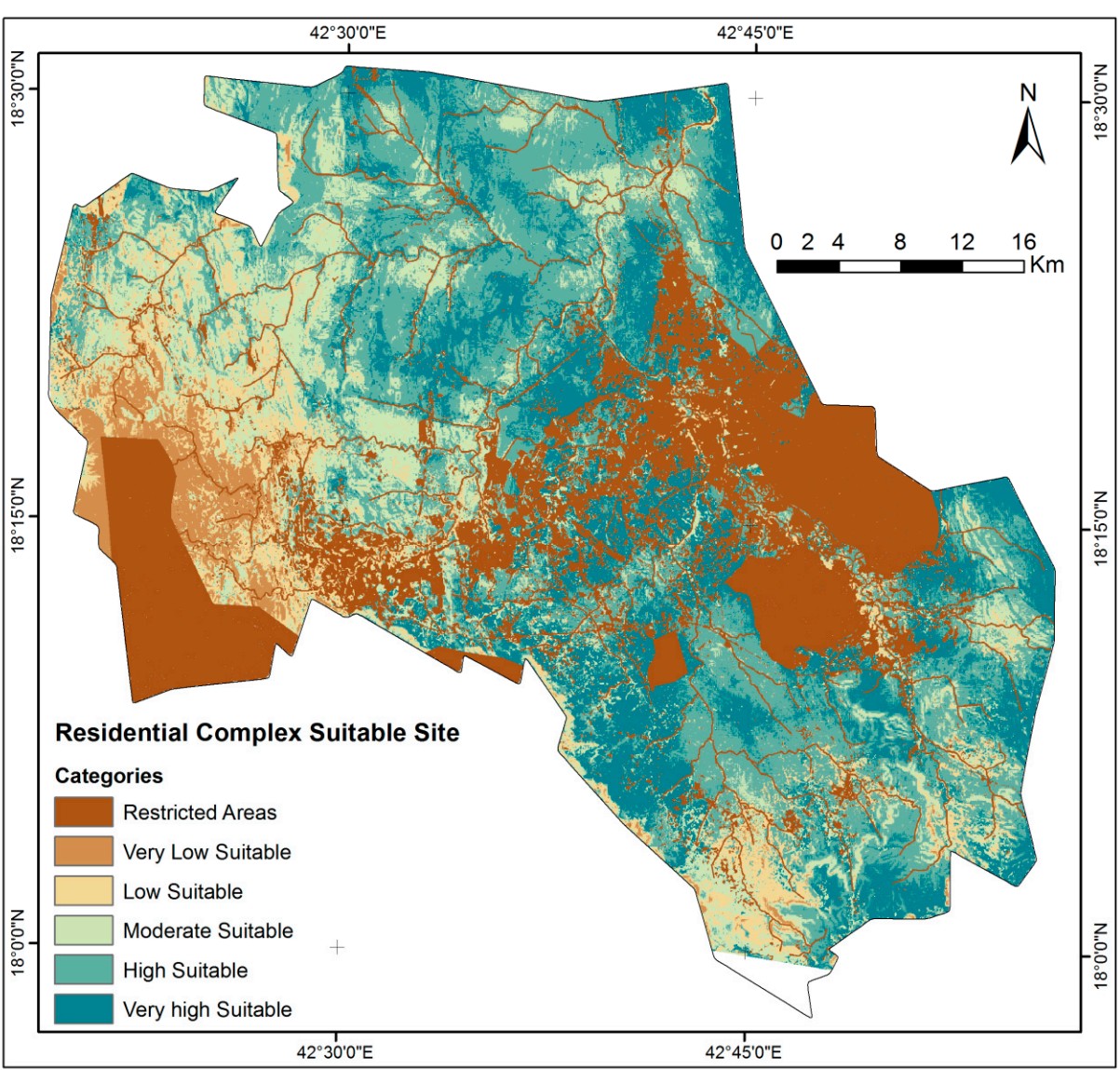

**Figure 7.** Classified residential complex building safe site map made using SOM.

### 3.3. Deeper Understanding of the Behavioral of Variables Using DNN-Based XAI

The present study employed a robust methodology involving DNN-based sensitivity analysis to gain a deeper understanding of the behavior of variables for constructing a new safe and eco-friendly residential complex in mountainous areas of Saudi Arabia. This analysis aids in identifying key factors influencing the selection of optimal locations, thereby contributing to the development of effective strategies for sustainable building complexes. The study utilized the "NeuralSens" package in R, leveraging a DNN model trained on a comprehensive dataset containing sixteen input variables and a target variable (RSSSI map). Regarding the training of the DNN model, Figure 8 demonstrates the optimization process, which was carried out using the "NeuralSens" package in R and took place over a span of 500 epochs. On the vertical axes, 'loss' and 'mean_absolute_error' are metrics used to evaluate the model's performance; loss is a calculation of the overall prediction error of the model (Figure 8, top panel), while mean_absolute_error measures the average magnitude of errors in a set of predictions without considering their direction (Figure 8, bottom panel). Typically, both metrics are minimized during training. In this case, as the epochs increase, a notable decrease in both metrics is observed, indicating effective learning and convergence of the model. The final values near zero suggest high accuracy in predicting the target variable (RSSSI map), with minimal average deviation from actual values.

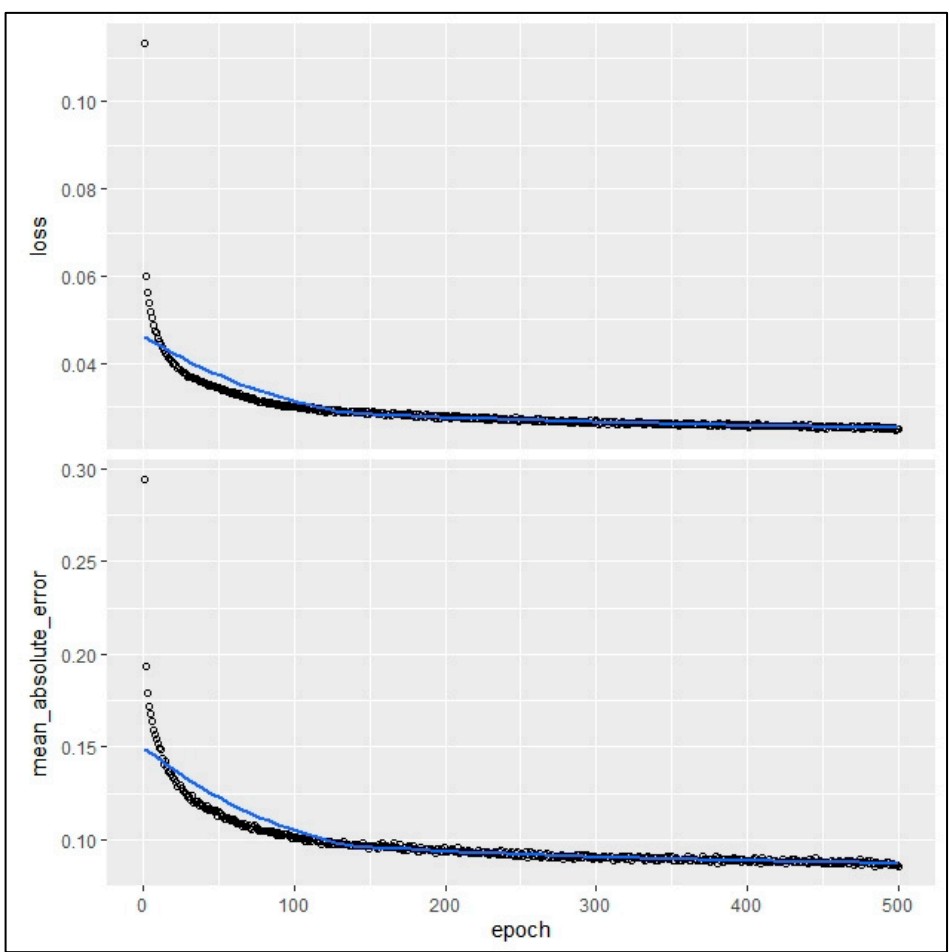

**Figure 8.** Training curves of a DNN model for finding optimal locations, depicting the loss (**top** panel) and MAE (**bottom** panel) decreasing and stabilizing over 500 epochs.

Upon model training, the weights of the DNN model were extracted and subjected to sensitivity analysis using the "SensAnalysisMLP" function from the "NeuralSens" package. The analysis provided insight into the impact of each input variable on the output variable (RSSSI model) in terms of mean and standard deviation. The sensitivity results were summarized and visualized through plots, while uncertainty computation was facilitated by computing sensitivity-based coefficients of variation (CVs). Higher CV values indicate greater uncertainty regarding a variable's influence on the output.

Behavioral Assessment of the Factors Influencing the Model

The sensitivity analysis depicted in Figures 9 and 10 elucidates the causal relationships between various environmental and infrastructural factors and their impact on selecting an optimal site for constructing a new safe and eco-friendly residential complex in Saudi Arabia's mountainous regions. Figure 9 (top panel) displays a scatter plot overlaying standard deviation (y-axis) against mean sensitivity values (x-axis) for various input variables affecting the selection of an optimal site for residential development. Each point represents an input variable, with its position indicating both the average influence it has on the model's predictions (mean sensitivity) and the variability or uncertainty of that influence (standard deviation). This plot helps in identifying the robustness of each variable's effect on the decision-making process. Variables clustered towards the right with low standard deviations are both influential and reliable in their impact, suggesting they are critical factors in the site selection criteria. Conversely, variables towards the left or with higher standard deviations indicate less influence or greater uncertainty, respectively. For instance,

variables like DEM and LULC are shown to be highly influential with low uncertainty, making them key factors in determining suitable locations for development.

The bar chart in Figure 9 (middle panel) quantitatively presents the mean sensitivity values of each input variable, offering insights into their relative influence on site suitability. Variables like LULC, the NDVI, and the DEM, with higher mean sensitivity and lower standard deviations, are indicative of stronger and more predictable influences on site selection. Figure 9 (bottom panel) displaying density plots, visualizes the distribution of the sensitivity scores for each input variable, enabling a better understanding of how changes in these variables can affect the suitability of a location. The undulations in the plots highlight the variability and uncertainty associated with each factor's influence, providing a nuanced view of how each contributes to decision making. This visualization aids in identifying variables that, despite having a strong mean influence, may exhibit significant uncertainty, impacting their reliability in the decision-making process. Figure 10 uses a beeswarm plot to depict the spread and concentration of sensitivity scores for each variable across a broader scale. The horizontal axis labeled 'sens' in both figures measures the sensitivity score, but while Figure 9 (bottom panel) focuses on a narrow interval, suggesting a detailed view within a small range, Figure 10 covers a wider range to highlight extreme values and outliers, providing a comprehensive overview of how input values influence model predictions across a broader spectrum. This contrast in scale between the two figures helps in interpreting the influence of each variable more dynamically, accounting for both typical and atypical conditions.

Therefore, this analysis reveals that variables with high mean sensitivity and lower variability, like the NDVI and DEM, significantly influence the selection of optimal locations for development due to their predictable impact on the environment. Conversely, variables like industrial presence, which show lower sensitivity and higher uncertainty, play a less decisive role, suggesting that areas with minimal industrial activities are preferable for reducing potential environmental conflicts. This comprehensive insight could guide strategic planning to enhance the safety and environmental integration of the housing developments within the natural landscape.

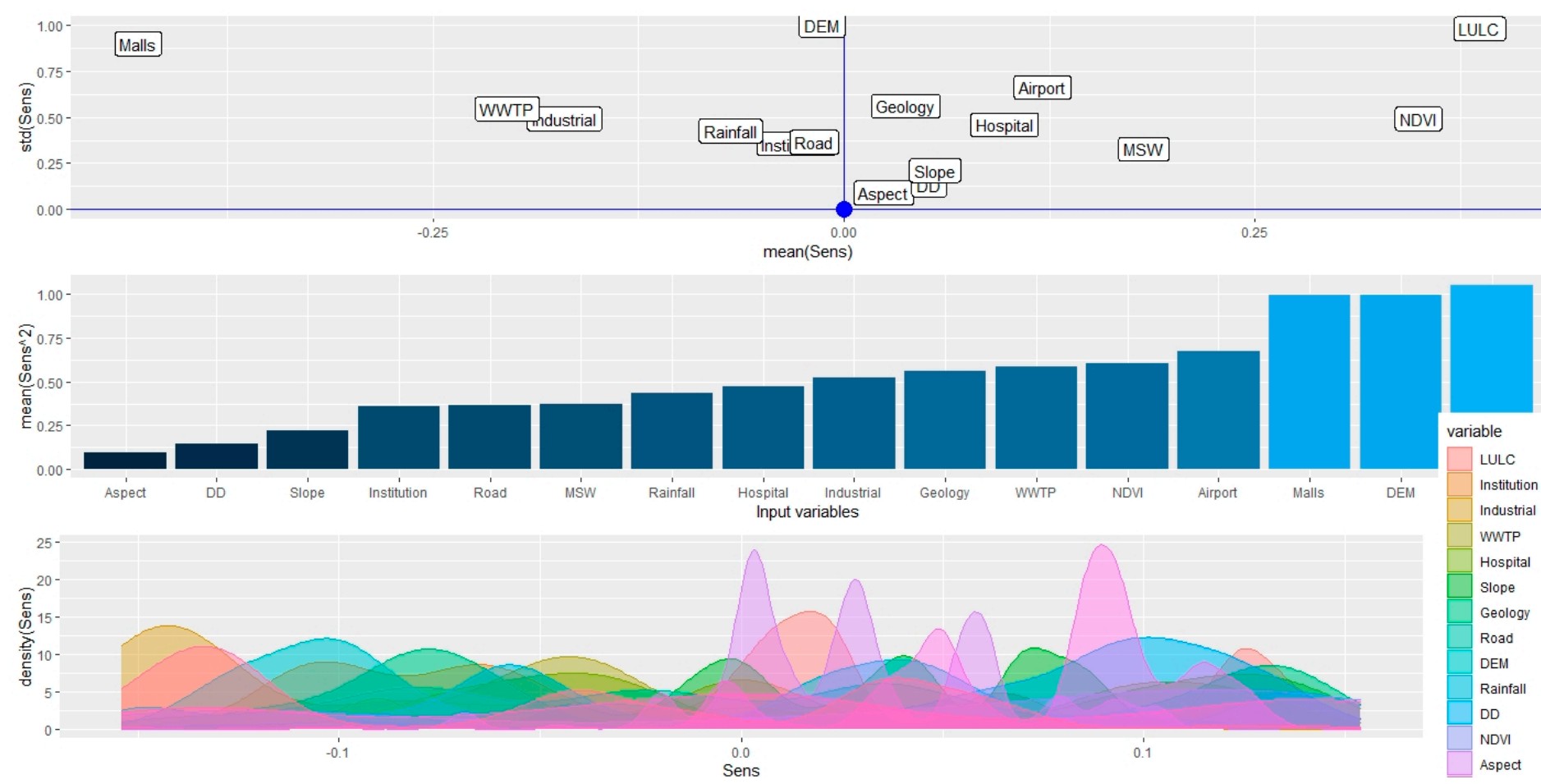

**Figure 9.** Sensitivity and uncertainty analysis for optimal location using Deep Neural Network-based sensitivity technique, depicting mean sensitivity across variables in a scatter plot (**top**), variable importance in a bar chart (**middle**), and the distribution of sensitivity scores in a density plot (**bottom**).

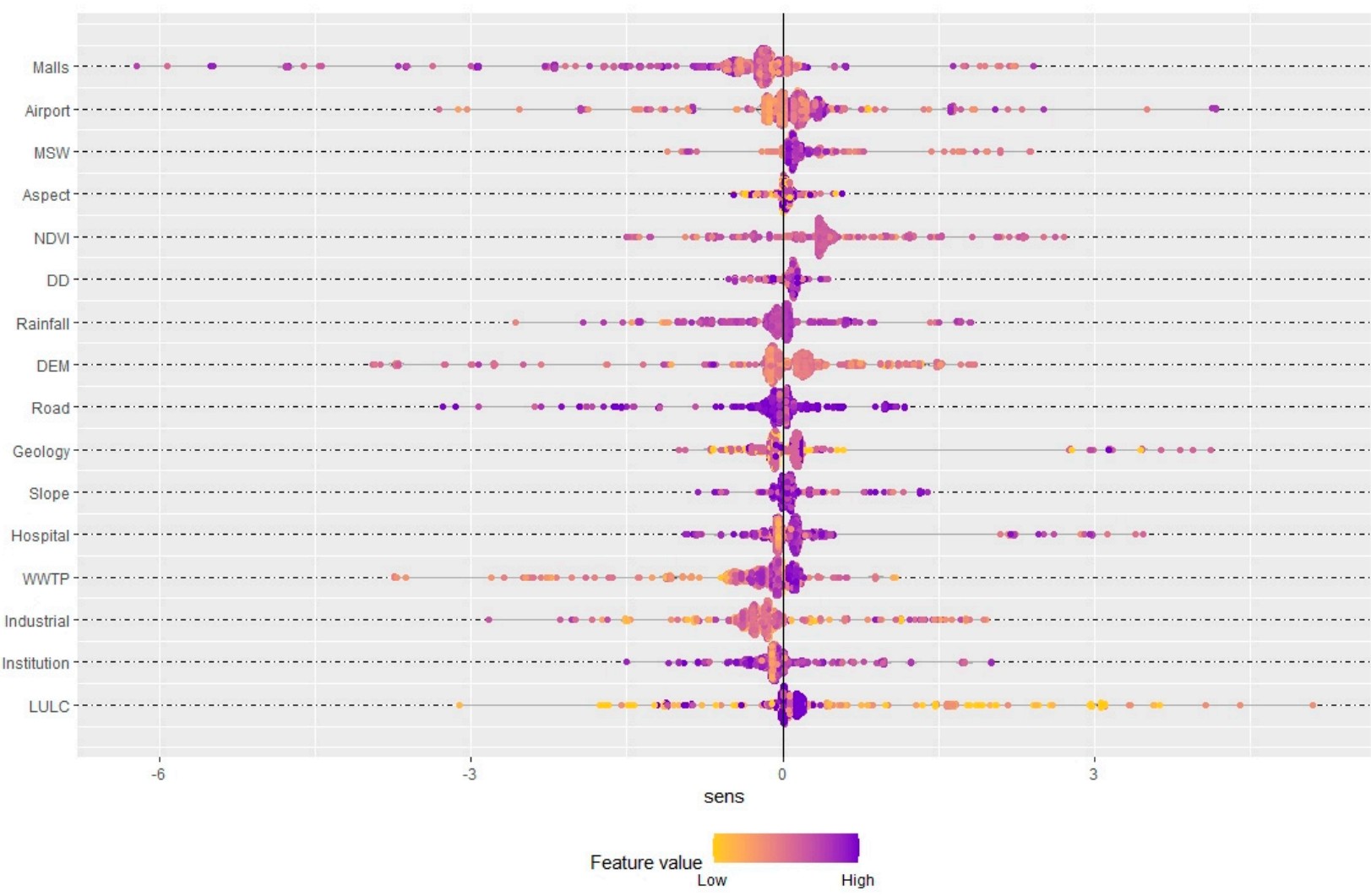

**Figure 10.** DNN-based SHAP value (game theory) for assessing the behavioral patterns of individual samples for optimal locations.

## 4. Discussion

The current study, focusing on Abha and the surrounding cities, shows how important it is to combine several sources of information in order to choose a safe location for the construction of buildings. The current guidelines do not emphasize any of the policies or procedures required for safe site selection in mountainous areas. Given the existing gap in knowledge, there arises a necessity to delineate a comprehensive set of variables aiding construction practitioners in pinpointing optimal sites for facilities and utilities within mountainous terrains [62]. This research embarked on a thorough exploration, primarily through a literature review, aimed at pinpointing three pivotal factors conventionally regarded in the processes involved in the selection of sites: geoenvironmental, geophysical, and socio-economic attributes. Subsequently, employing a survey methodology targeting construction experts affiliated with governmental and private entities operating within the Asir region of Saudi Arabia, a systematic inquiry unfolded. This investigation sought to meticulously evaluate these aforementioned crucial elements while also delving into additional factors influencing the designation of safe construction sites within mountainous landscapes. Responses were collected in five surveys sent to overseas professionals. These themes were created using both traditional data and RS (remote sensing) data. We applied a decision-making method using GIS software to develop a new building site using the Fuzzy-AHP model, as shown in Figure 6. Six categories of suitability maps were created: no development, very low suitability, medium suitability, high suitability, and very high suitability. While MCDA may at times introduce subjective elements into the criterion selection process and the assessment of their relative importance, the precision of outcomes is bolstered through the initial step of assigning weights to criteria, which is informed by expert insights and literature review. This foundational stage serves to instill objectivity and rigor into the decision-making process, thereby augmenting the accuracy and reliability of the final results.

The findings from our outdoor observations corroborated our hypotheses. Specifically, areas designated as highly and moderately suitable for development were identified as being free from any restrictions or protective designations and were deemed not susceptible to natural calamities. Interestingly, these zones were found in close proximity to pre-existing populated areas, major transportation routes, and educational and healthcare amenities. This underscores the importance of considering existing infrastructure and community resources in the assessment of suitable development zones. The results show that almost a third of the Abha–Khamis Mushait regions are used for unsuitable or less suitable purposes, comprising areas for crops (forests), protected areas, and agricultural land. This indicates that the use of the Fuzzy-AHP and GIS techniques will be of great help in preserving the environment and creating sustainable development plans in the future. This will significantly reduce the negative impacts on the ecological system and climate. The results also show that protected areas and geomorphological threats such as steep slopes, geological materials, and sudden flood risks limit the growth of the city. In this work, Fuzzy-AHP-MCDA was used based on geographical data to select a safe site for residential development. The aim of this study is to investigate how artificial values from pairwise comparisons are used by the system. In contrast, Duru et al. [63] delved into numerous studies on the Fuzzy-AHP, highlighting instances where the issue of matrix compatibility was not encountered despite the presence of decision inconsistencies. This underscores the complexity inherent in decision-making processes and the need for nuanced approaches to address such challenges effectively.

The incorporation of a DNN-based XAI approach significantly improves the evaluation of the suitability of potential new sites for a safe and environmentally friendly residential complex in mountainous regions of Saudi Arabia. By utilizing the complex learning capabilities of DNNs, this approach facilitates the complex analysis of the interactions of multiple variables and their collective impact on determining optimal site selection. This method enables decision makers to recognize the critical factors influencing site suitability and provides a comprehensive understanding of their relative contributions.

Consequently, DNN-based XAI not only helps identify areas with ideal conditions for complex development, such as areas with a higher vegetation density or efficient waste management infrastructure, but also highlights potential issues related to the presence of institutions, industrial activities, and other variables. This enables an informed decision-making process that ensures that the chosen location is in line with safety, sustainability, and environmental protection objectives.

Based on the above discussion, it can be concluded that for the construction of a safe and environmentally friendly building complex in sensitive mountain areas, several science-based strategies can be proposed to guide the development process. The analysis emphasizes the importance of variables such as the NDVI, MSW, and certain infrastructure aspects such as distance to airports and hospitals in determining the optimal location. To capitalize on these findings, one strategy could involve careful site planning that focuses on areas with higher NDVI values, indicating greater vegetation density and limiting deforestation. This approach not only improves the visual aesthetics of the complex, but also helps to improve air quality and ecological balance in the mountainous landscape. Given the moderate positive influence of municipal waste, it is essential to create an advanced waste management infrastructure that can accommodate the projected waste generation of the complex. This could include state-of-the-art recycling and disposal systems as well as initiatives to promote waste reduction and resource efficiency. In addition, the presence of hospitals and airports, both of which are associated with positive impacts, could be boosted for improved accessibility and emergency care to guarantee the safety and welfare of inhabitants. However, the study also emphasizes the uncertainties associated with certain variables such as institutions and industrial activities. Therefore, strategies need to prioritize areas with less institutions and industrial activity to mitigate potential conflicts and environmental problems. An integrated approach for land use planning, zoning regulations, and environmental impact assessments could help minimize negative impacts on the mountain ecosystem while promoting sustainable development.

However, our research is complemented by the findings of Ahmed et al., who discussed the application of artificial neural networks in the sustainable development of the construction industry and emphasized the central role of AI in improving practices in the industry [36]. Similarly, Xiang et al. highlight how AI can critically evaluate sustainability in green building engineering, emphasizing the integration of AI and energy considerations [37]. In addition, Pan and Zhang provide a critical overview of the role of AI in construction, pointing to future trends that our study anticipated and began to address through advanced computational modeling [38]. Venkatesh et al. discussed AI in smart city applications, reinforcing our use of sophisticated AI techniques for urban planning in a sensitive environmental context [39]. Furthermore, Kuhaneswaran et al. explore how the integration of geographic information systems with AI optimizes landfill site selection, which has parallels with our methodological approach for housing sites [40]. Waqar focuses on intelligent decision support systems using AI and machine learning in construction, which are similar to the decision support methods used in our study, highlighting the broader applicability and effectiveness of AI in this field [41]. The work of Seid et al. and Gao et al. on multi-agent systems in network management demonstrates the advanced capabilities of AI that could be integrated into our frameworks to improve dynamic and real-time decision-making processes in urban planning [42,43]. By utilizing these advanced technologies, our research not only addresses current needs but also creates a foundation for incorporating further AI advancements into urban development strategies to ensure environmentally sustainable and safe urban growth. These comparisons confirm the relevance and timeliness of our approach and indicate a substantial alignment with current research trends in the application of AI in various areas of urban development.

The future scope of this research is promising for developing more comprehensive strategies for the safe and environmentally friendly construction of building complexes in sensitive mountain regions. Further research could look at the dynamic interactions between variables over time, taking into account climatic changes and evolving land use

patterns. The incorporation of real-time data streams, remote sensing, and advanced predictive modeling techniques could refine the accuracy of site suitability assessments. In addition, the integration of renewable energy sources, green building technologies, and resilient design principles could further improve the environmental performance and resilience of complexes.

The study introduces an exhaustive and scientifically rigorous methodology for evaluating natural hazards concerning the selection of residential construction sites within mountainous terrains. This approach encompasses a thorough examination of various factors contributing to hazard assessment, aiming to enhance the understanding and mitigation of risks associated with residential development in such challenging landscapes. The use of advanced tools like the Fuzzy-AHP, GISs, and DNN-based XAI offers a new paradigm in urban planning and development, particularly in hazard-prone areas. The findings of this research not only contribute to the field of construction and urban planning but also pave the way for future studies that can further refine these methods and apply them in different contexts. As the study suggests, ongoing research in this area, especially considering the dynamic nature of environmental and socio-economic factors, is essential for developing more effective strategies for sustainable and safe construction in sensitive mountain regions.

## 5. Conclusions

This study presents a comprehensive and innovative framework for identifying optimal locations for safe and environmentally friendly residential complexes in the mountainous region of Abha and surrounding cities in Saudi Arabia. The study integrates geophysical, geoecological, and socio-economic parameters with traditional and advanced site selection methods to improve decision making. The results highlight the critical influence of topographic features such as slope and elevation, as well as vegetation density and rainfall patterns, using a Fuzzy-AHP weighting scheme. In addition, the implementation of DNN-based XAI provides deeper insights into the behavior of variables that significantly influence site selection. Despite the innovative contributions of the study, it does have limitations due to the uncertainties in the socio-economic parameters, suggesting a need for further refinement and the inclusion of additional variables such as air quality and socio-cultural factors. Future research could improve the accuracy of the model by incorporating more comprehensive predictive modeling and scenario analysis using advanced machine learning techniques. This research lays the foundation for future studies aimed at reconciling urbanization and environmental sustainability. It promotes interdisciplinary approaches that combine geospatial analysis, AI technologies, and socio-economic insights. This integration is crucial to making informed decisions that promote both human progress and environmental protection and contribute to a more resilient and sustainable urban future.

**Supplementary Materials:** The following supporting information can be downloaded at: https://www.mdpi.com/article/10.3390/su16104235/s1, Table S1: Land use/ land cover distribution (2020); Table S2: Matrix of pairwise comparisons for the 16 themes; Table S3: Fuzzy-AHP Priority; Table S4: Weights for thematic layers using Fuzzy-AHP techniques; Table S5: Weighted analysis.

**Author Contributions:** Conceptualization: D.A., A.M.A., S.T. and J.M.; data curation: A.M.A., S.T. and J.M.; formal analysis: D.A., A.M.A., S.T. and J.M.; funding acquisition: D.A.; methodology: D.A., A.M.A., S.T. and J.M.; project administration: J.M.; resources: A.M.A. and S.T.; software: A.M.A. and J.M.; validation: J.M., writing—original draft: D.A., A.M.A. and S.T.; writing—review and editing: J.M. All authors have read and agreed to the published version of the manuscript.

**Funding:** Funding for this research was given under award numbers R.G.P.2/287/44 by the Deanship of Scientific Research, King Khalid University, Ministry of Education, Kingdom of Saudi Arabia.

**Institutional Review Board Statement:** Not applicable.

**Data Availability Statement:** The datasets used and/or analyzed during the current study are available from the corresponding author on reasonable request.

**Acknowledgments:** The authors extend their appreciation to the Deanship of Scientific Research at King Khalid University for funding this work and supporting our research group under grant number R.G.P.2/287/44. The authors are also thankful to the USGS Earth Explorer for making the Landsat data freely available.

**Conflicts of Interest:** The authors declare no conflicts of interest.

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
