# Peer review of "Optimizing Residential Construction Site Selection in Mountainous Regions Using Geospatial Data and eXplainable AI"

_sustainability, doi:10.3390/su16104235_

Round 1
Reviewer 1 Report
Comments and Suggestions for Authors
Review#1: This paper proposes a novel methodology that combines socioeconomic parameters, fuzzy AHP, DNN-based Explainable Artificial Intelligence (XAI), and game theory. Furthermore, case studies reveal slope as the most crucial parameter and verify the effectiveness and superiority of the proposed approach. The following are my comments:
1. In the introduction, the authors should add some literature review on mathematical methods or artificial intelligent approaches for optimized, sustainable, and safe residential construction site selection.
2. As mentioned in the Introduction, there is still criticism of AHP's handling of uncertainties and inaccuracies. The author should point out what uncertainty exists.
3. Please carefully revise the grammar and format. The manuscript has some errors about this.
4. This paper is too lengthy and wordy. Please revise it.
5. Some figures are not clear, and the font should be revised to Times New Room. Please check it.
6. The authors should provide the detailed parameters of SOM in section 3.
7. The conclusion should be revised for improved clarity and conciseness.
8. The following research can be compared in terms of AI (not mine): 1:Multi-agent DRL for task offloading and resource allocation in multi-UAV enabled IoT edge network 2: A Two-stage Multi-agent Deep Reinforcement Learning Method for Urban Distribution Network Reconfiguration Considering Switch Contribution
Comments on the Quality of English Language
can be improved
Reviewer 2 Report
Comments and Suggestions for Authors
This study provides actionable insights for sustainable and safe residential development in Abha. It aids informed decision-making, balancing urban expansion with environmental conservation and hazard risk reduction. The study is well prepared, designed and conducted. The research is meaningful and innovative.
Other comments include:
1. The title is too long and makes it difficult to grasp the key points and core content. It is recommended to shorten the title.
2. This study is an interesting topic. Yet, the level of novelty of the contribution is not clear.
3. The paper dives into the technical details without providing a proper motivation, while the contextualization and motivation is fundamental.
4. There is a lack of quantitative analysis and comparative experiments of experimental results, and it is recommended to improve it.
5. "Table 1 Data used for the present study" appears twice.
6. There are some of the latest research papers on Sustainable Urbanization and Risk Assessment. It is recommended to read and cite (if applicable) what is helpful to the revision.
--Wee, S.J., Park, E., Alcantara, E. et al. Exploring Multi-Driver Influences on Indonesia's Biomass Fire Patterns from 2002 to 2019 through Geographically Weighted Regression
--Yamusa, I.B., Ismail, M.S. Futuristic Structural and Lithological Constraint Mapping of Landslides Using Structural Geology and Geospatial Techniques.
Comments on the Quality of English LanguageMinor editing of English language required.
Reviewer 3 Report
Comments and Suggestions for Authors
General assessment
This study examines the suitability of land for the construction of settlements in a defined study area in Saudi Arabia. The authors apply state of the art technology at its best level. The paper is undoubtedly worth publishing. Some minor improvements, outlined below, should be made prior to publication.
Methodology
The authors use both standard GIS location allocation and artificial neural network methodology. The GIS analysis results in a suitability map (Figure 7), the deep neural network analysis provides insights into the sensitivity of the input variables. This approach seems to be very beneficial, but the results of both methodologies are described more or less separately from each other, without establishing much mutual relationship. The relationship between the results of the two methods should be made more explicit.
One example. The description of the GIS method in section 2.3.12 Distance to Municipal Solid Waste states ‘… buffer zones were delineated using Euclidean distance measures, and the map generated was classified and evaluated utilizing the buffer distances as the criteria [37]. The second control point, situated at a considerable distance from the municipal solid waste disposal area, represents the most optimal choice for safe site selection for constructing residential complexes. In contrast, the first control point, positioned at a distance of 3000 meters, signifies a less suitable option for residential complex construction.’ When evaluating the results of the sensitivity analysis, it states: ‘Similarly, the amount of municipal solid waste generated in an area has a moderately positive influence, which emphasises the importance of infrastructure for waste management (line 596f). How do these two statements relate to each other?
The same comment can be made for most of the criteria used.
Specific and minor issues
Line 198 should probably read ‘ … north and west-facing slopes’ instead of ‘… north and east-facing slopes’
Figure 2, Part ‘Distance to Road Network’ obviously presents built up areas, but without explanation in the legend, which is confusing. Add an explanation in the legend or remove built up areas from the map.
The explanatory paragraphs of 2.3.8 Distance to Road Network and 2.3.9 Distance to Industrial Area are identical. Presumably the ,text of section 2.3.9 needs to be replaced with a correct version.
Subheading 2.3.10. should not be written in italics text style.
Line 255: distances farther than 6.000 meters are described as ‘unsuitable’. Does this mean that you are excluding all areas farther than 6.000 meters? (You do not, which is evident from the results.) A rewording of this sentence would be good.
2.3.12 Distance to Municipal Solid Waste: The greater the distance between residential neighborhoods and waste disposal facilities, the better for the wellbeing of residents, agreed. Having said that, good accessibility to these facilities is desirable for fast waste disposal. How do you handle this contradiction?
‘Figure 3’ should be replaced by ‘Figure 2 continued’ to be consistent with the numbering of the figures in the running text.
2.3.15 Distance to Waste Water Treatment Plant
Similar problem as in section 2.3.12: conflicting objectives between easy connection of residential neighborhoods to waste water treatment plants and the well-being of residents
Line 336ff: Where do we find variable x of Eq. 1 in Figure 3? Horizontal axis? Please insert x in Figure 3.
Line 406, end of line: Wi i is the …: ‘i’ seems to need to be removed
Line 414: it seems that RSSSZ must be replaced by RSSPI. If not, RSSSZ needs to be explained.
Figure 8, better explanation needed: add explanations for the vertical axis units ‘loss’ and ‘mean_absolute_error’. Add an interpretation of the vertical axis values. What do they mean?
Figure(s) 9: add explanations for each horizontal and vertical axis. What is the benefit of presenting the bar chart of ‘Input variables’? For me, the message of Figure 9c is unclear. How are the presented undulations related to the input variables?
Figures 9c and 10 have the same labeling of the horizontal axis ‘Sens’. If this is correct, than how are both graphs related? Figure 9c covers an interval of approximately [-0.15, +0,15] on the horizontal axis, Figure 10 covers an interval of [-6, +6].
